# Research

evolution, genetics, immunology

sex-ratio, inversion, infection, susceptible, segregation distorter

**Author for correspondence:**
Robert L. Unckless
e-mail: unckless@ku.edu

One contribution to the Special Feature 'Natural and synthetic gene drive systems'. Guest edited by Nina Wedell, Anna Lindholm and Tom Price.

# An assessment of the immune costs associated with meiotic drive elements in *Drosophila*

Jenna Kay Lea and Robert L. Unckless

Department of Molecular Biosciences, University of Kansas, Lawrence, KS 66045, USA

RLU, 0000-0001-8586-7137

Most organisms are constantly adapting to pathogens and parasites that exploit their host for their own benefit. Less studied, but perhaps more ubiquitous, are intragenomic parasites or selfish genetic elements. These include transposable elements, selfish B chromosomes and meiotic drivers that promote their own replication without regard to fitness effects on hosts. Therefore, intragenomic parasites are also a constant evolutionary pressure on hosts. Gamete-killing meiotic drive elements are often associated with large chromosomal inversions that reduce recombination between the drive and wild-type chromosomes. This reduced recombination is thought to reduce the efficacy of selection on the drive chromosome and allow for the accumulation of deleterious mutations. We tested whether gamete-killing meiotic drive chromosomes were associated with reduced immune defence against two bacterial pathogens in three species of *Drosophila*. We found little evidence of reduced immune defence in lines with meiotic drive. One line carrying the *Drosophila melanogaster* autosomal Segregation Distorter did show reduced defence, but we were unable to attribute that reduced defence to either genotype or immune gene expression differences. Our results suggest that though gamete-killing meiotic drive chromosomes probably accumulate deleterious mutations, those mutations do not result in reduced capacity for immune defence.

## 1. Background

Organisms must adapt to affronts that are both extrinsic (changing habitat, temperature, predators, parasites) and intrinsic (selfish genetic elements such as transposable elements, selfish B chromosomes and meiotic drive elements). When adapting to factors that have the capacity for genetic adaptation themselves, the interaction can involve a coevolutionary arms race where both host and adversary rapidly and recurrently adapt to the co-adapting partner [1,2]. Two strong selective pressures in many species are pathogens and selfish genetic elements, such as transposable elements and meiotic drivers, and it is plausible that the presence of selfish genetic elements might impede immune defence.

Meiotic drive is broadly defined as any process where one chromosome or part of a chromosome biases its own transmission during gametogenesis at a cost to the homologous chromosome [3]. There are several documented cases of true meiotic drive, where one genotype preferentially makes it into the egg during oogenesis and the other genotype is preferentially relegated to the polar bodies, preventing inheritance in the next generation [4–6]. However, more commonly documented are the gamete killers where, during spermatogenesis, one genotype kills or disables the sperm carrying the homologous chromosome [7,8]. Such gamete-killing meiotic drivers may occur on autosomes, but most documented cases are sex-linked, probably due to the striking resulting phenotype—males carrying the driving X chromosome sire mostly daughters. In only a handful of cases do we understand the genetic

basis of drive; however, most drive loci in multicellular organisms are associated with chromosomal inversions or regions of otherwise suppressed recombination [3]. Reduced recombination is beneficial for the driver because it restricts recombination with the target locus on the homologous chromosome, which prevents formation of resistant non-driving chromosomes and suicidal driving chromosomes. Reduced recombination also allows the drive locus to recruit linked enhancer loci that may strengthen or stabilize drive [9]. However, from the whole-organism perspective, this reduced recombination may lead to the accumulation of deleterious mutations. These cannot be purged through recombination and purifying selection but alone are not strong enough to outweigh the selfish benefit the driver provides to the chromosome [10,11].

Due to the evolutionary pressure imposed by pathogens, genes involved in immune defence are among the fastest evolving genes in the genome for several groups of organisms [12–16]. This is especially true in many invertebrate species, including *Drosophila*. Invertebrates lack the canonical vertebrate adaptive immune system so rely entirely on their innate immune system to combat infection. Insect innate immunity involves antiviral RNAi, cellular (encapsulation, melanization, phagocytosis and coagulation) and humoral immunity (secretion of immune effectors such as antimicrobial peptides) [17]. Often, antimicrobial peptide (AMP) induction is used as a proxy for the magnitude and speed of the immune response. In *Drosophila*, genes involved in the immune response are distributed throughout the genome, but antimicrobial peptides are overrepresented on the second chromosome and completely missing from the sex chromosomes.

The notion that selfish genetic elements could influence the immune response is not new—there are several examples of insertions of transposable elements interfering with immune defence [18–20]. Probably the most well-known example is the cooption of transposable elements to facilitate V(D)J recombination and produce diverse antibodies and receptors in lymphocytes in vertebrates. Often, transposable element insertions function by influencing gene expression in *cis* [21]. For example, an endogenous retrovirus insertion upstream of the human *AIM2* gene influences expression and subsequently the inflammatory response [22]. In contrast, in *Drosophila melanogaster*, a transposable element insertion in the protein-coding region of the CHK kinase gene, *CHKov1*, results in increased resistance to an RNA virus [23].

Here, we test whether meiotic drive elements are associated with changes in immune defence. We hypothesize that the mechanism relies on the accumulation of mutations deleterious to immune defence on drive chromosomes. We test three *Drosophila* meiotic drive systems: two are sex-ratio meiotic drive and one is autosomal (second chromosome). The sex-ratio meiotic drive element in *Drosophila affinis* causes males to sire about 98% daughters, and is found at between 2% and 10% in populations across the eastern United States [24]. The drive element is associated with two large inversions on the X chromosome [25]. The sex-ratio meiotic drive element in *Drosophila neotestacea* causes males to sire greater than 95% daughters and is found at varying frequencies in populations across mid- to northern North America [26–28]. The drive element is also associated with inversions and is likely caused by a duplication of the gene encoding the nuclear import factor, *importin-a2* [29]. Finally, the autosomal driver in *D. melanogaster*, *Segregation distorter*

(*SD*), occurs at varying frequencies in populations worldwide but arose in African populations [9,30,31]. It is located on the second chromosome and is associated with inversions that reduce recombination and link *SD* driver, enhancer and responder loci over large parts of the chromosome. For each species, after controlling for genetic background, we infected individuals with and without the driver (heterozygous or homozygous) and monitored survival for five to seven days. We find that meiotic drive chromosomes are not generally associated with lower survival after infection, but that one particular *SD* chromosome is more susceptible. For this chromosome, we explored whether the increased susceptibility was associated with a global reduction in the speed or magnitude of induction of the immune system or because of mutations in a particular AMP known to be associated with survival to one of the pathogens employed.

## 2. Methods

### (a) *Drosophila* husbandry

Maintaining sex-ratio lines requires a sex-ratio (SR) stock and a standard/wild-type (ST) stock. Because SR males produce only (or mostly) daughters, SR lines are maintained by (i) crossing SR/SR homozygous females to SR/Y males to produce SR/SR homozygous females, (ii) crossing SR/SR homozygous females to ST/Y males to produce SR/Y males and (iii) ST/ST females to ST/Y males to maintain the ST stock. This process is then repeated in perpetuity, meaning that the SR and ST lines should be isogenic except for the X chromosome.

*Drosophila neotestacea* SR and ST lines were obtained from Dr Kelly Dyer (University of Georgia) and maintained in vials with instant *Drosophila* medium (Formula 4-24, Carolina Biological Supply Company, Burlington, NC), a dental roll and a chunk of commercially available white button mushrooms. However, after infection, we omitted the mushroom and added an additional cotton roll to stabilize humidity. Flies were maintained at 22°C on a 12 h light/dark cycle and were allowed to mate 1–2 days, then separated by sex and aged for 5–10 days before infection. For *D. neotestacea* SR drive, we infected three female genotypes (SR/SR, SR/ST and ST/ST) and two male genotypes (SR/Y and ST/Y).

*Drosophila affinis* lines were collected by Rob Unckless and John Jaenike and described in Unckless *et al*. [24]. They were maintained on malt media: 10 g agar, 60 g semolina flour, 20 g yeast and 80 g malt extract in 1 l water (supplemented with Tegosept and propionic acid) [32]. *Drosophila affinis* flies were maintained at 20°C on a 12-h light/dark cycle but crossing and pre-infection maintenance were otherwise equivalent to *D. neotestacea*. For *D. affinis* SR drive, we infected two female genotypes: SR/SR and ST/ST, and two male genotypes: SR/Y and ST/Y.

Autosomal *Drosophila melanogaster Segregation distorter* lines were obtained from Dr Amanda Larracuente and were segregating on variable genetic backgrounds. We used two lines with multiple balancers to move the *SD* second chromosomes onto an otherwise A4 [33,34] background (electronic supplementary material, figure S1). As a control, we also moved four non-driving second chromosomes from African inbred lines and another laboratory stock [35]. Some second chromosomes are homozygous lethal, so we ended up with the following set of second chromosomes all on an otherwise A4 background: *SDFR43*, *SDMad*, *SD72/Cyo*, *Gla/Cyo*, *Zi104*, *Zi31N* and *Zi335*.

### (b) Survival assays

Flies were infected with *Providencia rettgeri* (Dmel) and *Enterococcus faecalis* (BPL), two natural pathogens of *Drosophila* that cause intermediate mortality at relatively low doses [36,37]. Infections

were carried out by (i) growing bacteria in LB (*P. rettgeri*) or Todd Hewitt (TH, *E. faecalis*) broth overnight to stationary phase and diluting to the appropriate optical density ($OD_{600} = 1.0$ for *P. rettgeri* and $OD_{600} = 1.5$ for *E. faecalis*), (ii) pricking flies in the thorax with a tungsten minutien pin dipped in the bacteria suspension and (iii) monitoring survival daily for 5–7 days [38]. Flies were housed in groups of 10 after infection. All flies were infected between 09.00 and 14.00 CST, and once infected, vials were placed in an incubator at temperatures listed above with approximately 80% humidity. Additionally, the genotype, infector, treatment and time of infection were recorded. We assessed differences in fly survival using a mixed-effects Cox proportional hazard test implemented in R [39,40]. For sex-ratio meiotic drive species, we either considered sex together with the number of SR chromosomes nested in sex or considered each sex separately. For the *D. melanogaster* lines, we considered those over a balancer and those homozygous either altogether or in separate models. In each case, we considered genotype or drive chromosome count and block as fixed effects with vial as a random effect.

## (c) Genotyping for *Diptericin*

Natural genetic variation in the antimicrobial peptide, *diptericin*, in *D. melanogaster* has significant influence on survival after infection with *P. rettgeri* specifically [41,42]. Since *diptericin* is on the second chromosome along with the *SD* locus, we genotyped all *D. melanogaster* lines for *diptericin* by Sanger sequencing using previously published primers [43]. PCR and sequencing were performed using standard techniques and sequences were aligned in GENEIOUS v. 10.2.6 (Auckland, New Zealand). Genbank accession numbers MN431479-MN431485.

## (d) Measuring the induction of immune gene expression using quantitative PCR

We used heat-killed bacteria to measure induction of expression after exposure because it avoids the confounding effects of different bacterial proliferation rates in different genotypes, which could then also alter the expression profiles of that host. Bacteria were heat-killed by incubating at 65°C for 35 min.

We measured *D. melanogaster* immune gene expression at zero and eight hours post exposure in *SD* and non-*SD* lines to determine whether differences in survival after infection could be predicted by the baseline or induced expression of genes involved in the immune response. We chose to measure expression of two antimicrobial peptides: *Diptericin*, which is located within one of the *SD*-associated inversions, and *Cecropin B*, which is on the third chromosome and therefore unlinked to *SD*.

We exposed flies to heat-killed bacteria using the same needle-prick procedure as described for live bacteria described above (but note that we use exposure instead of infection). For qPCR, we flash froze flies in three groups of three at 0 and 8 hours post exposure. Once collected, RNA isolations and cDNA library preps were performed on each of the trios to prepare for qPCR. Frozen fly tissue was homogenized by autoclaved motor pestle and RNA was extracted using the Tissue and Tough-to-Lyse Samples protocol from the Quick-RNA MicroPrep Kit (Zymo Research, Irvine, CA). After extractions, RNA was quantified using a DS-11 FX+ Spectrophotometer/Fluorometer (DeNovix, Wilmington, Delaware). Prior to cDNA preparation, all RNA was diluted to the lowest concentration sample in RNase/DNase-free H20. iScript cDNA Synthesis Kit protocol (Bio-Rad, Hercules, CA) modified for half volume reactions (2 µl 5× iScript Reaction Mix, 0.5 µl iScript Reverse Transcriptase, 7.5 µl diluted template) were used to complete cDNA synthesis in a 96-well plate. Finally, Hard-Shell 96-Well PCR Plates (Bio-Rad, Hercules, CA) were prepared for qPCR following SsoAdvanced Universal SYBR Green Supermix (Bio-Rad,

Hercules, CA) RNA protocol. The primers used in the master mixes prepared before adding cDNA template were Dpt-PP33592_F (ACGCCACGAGATTGGACTG), Dpt-PP33592_R (CAGCTCGGTTCTGAGTTGC), CecB-PD41776_F (CTGGGAAACTCAGAGGCTGG) CecB-PD41776_R (CCTGGATTGAGGCATCCCTG), rp49- PD41810_F (AGCATACAGGCCC-AAGATCG), and rp49- PD41810_R (TGTTGTCGATACCCTT-GGGC) (DRSC FlyPrimerBank, Boston, MA). Once all reagents were added, plates were sealed with Microseal 'B' Seals (Bio-Rad, Hercules, CA) and run under CFX Connect thermal cycling protocol (Bio-Rad, Hercules, CA). We employed standard Bio-Rad cycling conditions (95°C for 3 min, 40 cycles of 95°C for 10 s then 55°C for 30 s, 95°C for 10 s then a melt curve from 65°C to 95°C at 0.5°C increments every 5 s). qPCR data was analysed by analysis of variance with target gene critical threshold cycle as the dependent variable and control gene (rp49) critical threshold cycle and genotype as independent variables.

# 3. Results

## (a) Sex-ratio meiotic drive chromosomes are not associated with increased susceptibility to infection

We tested whether sex-ratio meiotic drive X chromosomes were associated with increased susceptibility to infection using two divergent *Drosophila* species (*D. neotestacea* and *D. affinis*). Note that while *D. neotestacea* carries the ancestral X chromosome (Muller A), an ancestor of *D. affinis* experienced a fusion of Muller elements A and D (*D. melanogaster* chromosomes X and 3 L, respectively) [44,45]. Therefore, approximately 40% of the *D. affinis* genome is X-linked.

*Drosophila neotestacea* females carrying at least one SR chromosome were actually less susceptible to infection with *E. faecalis* than homozygous wild-type females (Cox proportional hazard $p_{SR/ST} = 0.0055$, $p_{SR/SR} = 0.0021$; figure 1*a*; electronic supplementary material, tables S1 and S2). Interestingly, SR/SR females are more susceptible to the Gram-negative *P. rettgeri* than either SR/ST or ST/ST—though not significantly so (figure 1*b*; electronic supplementary material, table S1; $p_{SR/SR} = 0.13$). SR male *D. neotestacea* displayed no difference in susceptibility compared to ST males (*E. faecalis* $p_{SR} = 0.56$, *P. rettgeri* $p_{SR} = 0.93$). We also used a model incorporating both sexes and SR chromosome copy number and found that while sex was not a significant predictor of survival, copy number nested in sex (for females) was significant for *E. faecalis* ($p_{sex(female):copy} = 0.004$) and marginal for *P. rettgeri* ($p_{sex(female):copy} = 0.064$)—both results mirroring what was found for females considered alone.

We found no significant differences in susceptibility to infection with either *E. faecalis* or *P. rettgeri* in *D. affinis* females (figure 1*c,d*; electronic supplementary material, tables S1 and S2; *E. faecalis* $p_{SR/SR} = 0.63$, *P. rettgeri* $p_{SR/SR} = 0.46$) or males (*E. faecalis* $p_{SR/SR} = 0.27$, *P. rettgeri* $p_{SR/SR} = 0.32$). However, in the model incorporating both sexes and accounting for SR chromosome copy number, sex was a significant predictor of survival for both pathogens (*E. faecalis* $p_{sex} = 0.044$, *P. rettgeri* $p_{SR/SR} = 0.005$). Interestingly, females were more susceptible to *P. rettgeri*, while males were more susceptible to *E. faecalis*.

## (b) One *SD* chromosome in *Drosophila melanogaster* is associated with higher susceptibility to infection

For the autosomal meiotic driver, *SD*, in *D. melanogaster*, we considered three different *SD* chromosomes isolated in

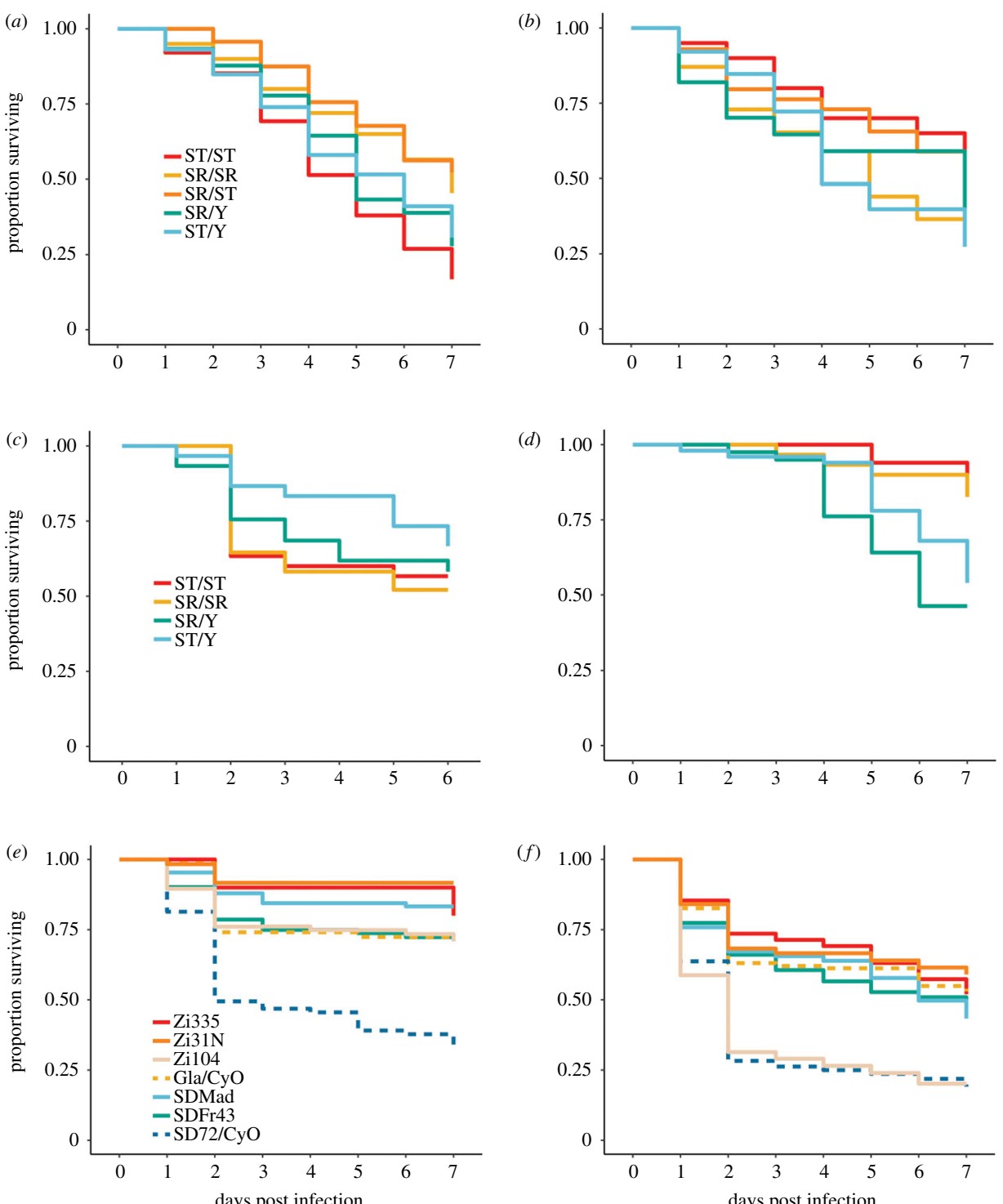

**Figure 1.** Survival of lines with and without meiotic drive chromosomes after infection. (*a,b*) *D. neotestacea*, (*c,d*) *D. affinis*, (*e,f*) *D. melanogaster*. (*a,c,e*) Infection with *E. faecalis*, (*b,c,f*) infection with *P. rettgeri*. (Online version in colour.)

different parts of the world and that exhibit different inversion arrangements. In addition, one line, *SD72*, contains mutations that render it homozygous lethal and therefore it was maintained over a second chromosome balancer (*CyO*), as was its matched control line. We tested for the effect of *SD* separately in the hemizygous, balanced state and in homozygotes where viable. The *SD72/Cyo* line was significantly more susceptible to infection than the *Gla/CyO* matched control for both *E. faecalis* ($p_{Sd72} < 0.001$) and *P. rettgeri* ($p_{Sd72} < 0.001$). In fact, for *E. faecalis* infections, *SD72/CyO* was a clear outlier for susceptibility among all lines (figure 1*e*). *SD* lines were not significantly more susceptible to infection with either pathogen when homozygous

($p > 0.3$ for all comparisons). One of the wild-type lines, Zi104, showed similar susceptibility to *P. rettgeri* as *SD72*. It is possible that the susceptibility to *P. rettgeri* in one or both of these lines is due to genetic variation in the antimicrobial peptide, *Diptericin*, which segregates for both null alleles and nonsynonymous SNPs that significantly influence susceptibility to specifically *P. rettgeri* [41,42].

## (c) *Diptericin* genotype does not explain the increased susceptibility to *Providencia rettgeri* in *SD72* and *Zi104*

To determine whether the increased susceptibility to infection with *P. rettgeri* might be due to *Diptericin* genotype, we

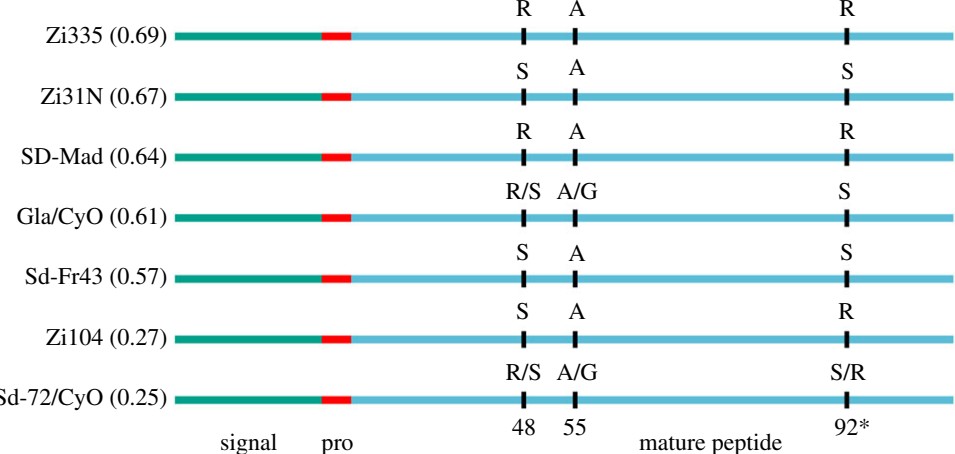

**Figure 2.** Diptericin genotype does not predict infection susceptibility to *P. rettgeri* in *D. melanogaster* lines employed. Each line, proportion surviving *P. rettgeri* infection four days after infection in parentheses (in decreasing order), and Diptericin genotype are displayed. Signal peptide, pro-piece and mature peptide are noted in different colours and three amino acid polymorphisms are displayed (positions 48, 55 and 92). An arginine (R) at residue 92 (denoted with an asterisk) is associated with increased susceptibility to *P. rettgeri* in other studies. (Online version in colour.)

sequenced the *Diptericin* gene for all seven lines studied. Alleles that could affect susceptibility include several null alleles (segregating at nearly 20% frequency in some populations) and a serine/arginine polymorphism at residue 92 of the full protein that is associated with a 40% drop in survival after infection [42]. We found no evidence for segregating null alleles among the seven lines, but four lines carried the susceptible arginine allele at residue 92 (figure 2). Surprisingly, these lines were not generally more susceptible than other lines, but both lines with very low survival after infection had at least one arginine allele. Thus, the Diptericin genotype does not seem to be strongly associated with susceptibility to *P. rettgeri* in this study.

### (d) Differences in antimicrobial peptide expression among lines do not explain variation in susceptibility

A common readout of the immune system is induction of antimicrobial peptide expression after infection. We measured relative expression of two AMPs at zero and eight hours after exposure to heat-killed *P. rettgeri* to determine whether either baseline or induced expression differed between susceptible and resistant genotypes. The heat-killed bacteria ensured that differences in expression observed were not due to inconsistent bacterial loads experienced because of differences in immune defence. Heat-killed bacteria generate a robust, but short-lived immune response [46]. The susceptible genotypes (*SD72/CyO* and *Zi104*) did *not* have reduced expression of *Diptericin* or *Cecropin* at either 0 or 8 h post exposure to heat-killed bacteria. In fact, second chromosome genotype does not significantly influence expression directly after exposure to heat-killed *P. rettgeri* (figure 3; electronic supplementary material, tables S3 and S4; $p = 0.119$). At 8 h post exposure, genotype is associated with expression ($p = 0.0001$), but the susceptible genotypes ranked third and fifth (out of seven) for highest expression. The same pattern holds for expression of *Cecropin*: there is no effect of genotype at zero hours ($p = 0.119$) and at 8 h post exposure genotype is associated with expression ($p < 0.0001$), but this does not predict survival. Therefore, the susceptibility of the *SD72/CyO* line is not explained

either by *Diptericin* genotype or AMP expression during early infection.

## 4. Discussion

Pathogens and selfish genetic elements impose unique challenges on their hosts. Their capacity to coevolve requires recurrent bouts of adaptation from the host just to keep pace. Gamete-killing meiotic drivers are often associated with large chromosomal inversions and the lack of recombination on drive chromosomes resulting from these inversions is thought to lead to a build-up of deleterious mutations. We tested whether one life-history trait— immune defence—is compromised when individuals carry a meiotic drive chromosome. In three species (two sex-ratio meiotic drive systems and one autosomal meiotic drive system), we found only one drive chromosome that was consistently associated with increased susceptibility to bacterial infection. This susceptibility did not appear to be associated with *Diptericin* genotype—a key determinant of *P. rettgeri* infection outcomes in other studies—nor was there evidence that the drive line with decreased survival was less able to induce an immune response compared to other lines.

We examined only a single sex-ratio meiotic drive chromosome each for *D. affinis* and *D. neotestacea*, but we examined three different *SD* chromosomes in *D. melanogaster*. Interestingly, we see considerable variation within *SD* chromosomes for susceptibility to drive. Older drive chromosomes likely segregate for several different driver haplotypes, each accumulating its own suite of enhancers and deleterious mutations.

The accumulation of deleterious mutations on a meiotic drive chromosome might lead to reduced immune defence in two distinct ways. First, deleterious mutations in key immune genes in or near drive-associated inversions could accumulate because reduced recombination prevents them from being purged. The process of the accumulation of deleterious mutations in the absence of recombination is known as Muller's ratchet [47] and may be particularly severe when the genes involved are only necessary conditionally (i.e. during infection). Put differently, mutations that reduce immune defence are only really deleterious during infection. If infection is rare, the joint effects of relaxed constraint due to

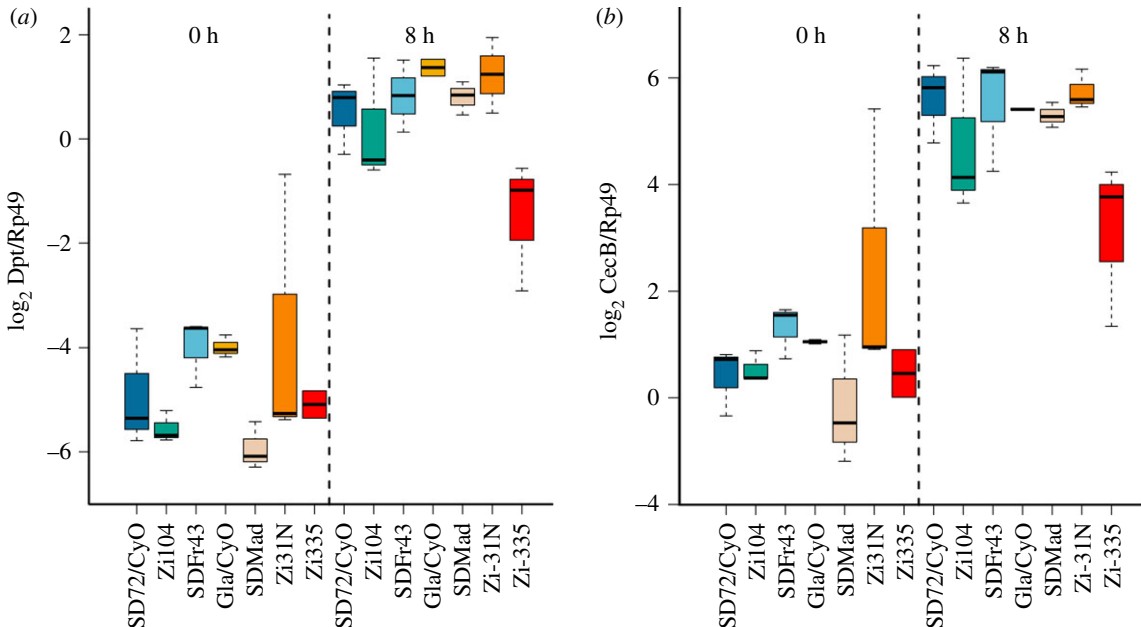

**Figure 3.** AMP expression is not associated with susceptibility to infection with *P. rettgeri*. Panels show relative expression (log$_2$ AMP/control) of (*a*) *Diptericin* and (*b*) *Cecropin B* at 0 and 8 h after a needle prick to the thorax. The needle was dipped in heat-killed *P. rettgeri*. Genotypes are ordered by proportion alive at day 4 post infection from experiments in figure 1. Boxplots represent the median value (thick black line), 25th and 75th quantiles (box) and extreme values (whiskers). (Online version in colour.)

conditional fitness costs and reduced ability to purge deleterious mutations might exacerbate Muller's ratchet. This is somewhat analogous to the role that epistasis plays in Muller's ratchet—where fitness costs are conditional upon genetic background, not infection—though the conclusions about the effects of epistasis are conflicting [48,49].

The second way that immune defence could be compromised because of drive chromosomes is through *trans* effects on immune gene expression. This could be because upstream modulators of the immune response are associated with the drive inversions or because drive chromosomes alter chromatin landscape genome-wide, leading to differential expression. There is some evidence that heterochromatin in *Drosophila* influences expression of genes in several key pathways including immunity [50–53].

We hypothesized that if autosomal *SD* in *D. melanogaster* was associated with increased susceptibility to infection, it would be due to the accumulation of deleterious mutations in immune genes. In contrast, if sex-ratio meiotic drive chromosomes were associated with increased susceptibility, it would be due to *trans* effects on gene expression. There is a general paucity of immune genes on the X chromosome (Muller element A) in *D. melanogaster* and an excess (especially of immune effectors) on the second chromosome (Muller elements B and C): chromosome 2—111 immune of 7005 total, chromosome 3—75 immune of 7268 total, chromosome X—14 immune out of 2612 total (1.58%, 1.03% and 0.05% respectively, Fisher Exact Test $p < 0.001$). In an ancestor of *D. affinis*, the X chromosome fused with an autosome leading to a large X chromosome (40% of the genome, Muller elements A and D) [45]. In *D. neotestacea*, the X chromosome is mostly syntenic to that in *D. melanogaster* [44]. Thus, there are relatively fewer immune genes for deleterious mutations to perturb linked to the sex-ratio meiotic drive systems compared to the autosomal drive system. However, we found no evidence that the susceptible *SD* line in *D. melanogaster* either had lower baseline AMP expression levels or induced AMP expression levels.

So, why don't we see a more pronounced influence of meiotic drive chromosomes on immune defence? One possibility is that the drive chromosome inversion haplotype is often common enough in populations to slow Muller's ratchet. An autosomal driver at an equilibrium frequency of 10% would find itself in homozygotes 1% of the time and deleterious mutations could be purged from the drive chromosome. However, there are ample examples of drive chromosomes that, when homozygous, render the carrier inviable or sterile [3,8,26]. So either deleterious mutations must arise on these chromosomes or the drivers themselves could cause the inviability or sterility. Another possibility is that drive chromosomes turn over rapidly enough that Muller's ratchet does not have time to accumulate many deleterious alleles [30]. Perhaps this explains why in *D. melanogaster*, the older *SD72* was more susceptible but the newer *SDFR43* was not [54]. Overall, meiotic drive chromosomes did not have a pronounced influence on immune defence. It would be interesting to determine whether other life-history traits would show evidence for the accumulation of deleterious mutations on drive chromosomes [55] or whether the sterility and inviability often observed in conjunction with drive chromosomes is actually due to a simple genetic change that is integral to the drive machinery itself.

Data accessibility. Raw data and analysis scripts are either presented in the electronic supplemental material or is available from the Dryad Digital Repository: https://doi.org/10.5061/dryad.26sm738 [56].

Authors' contributions. J.K.L. and R.L.U. conceived the experiments, analysed the data and wrote the manuscript. J.K.L. performed the experiments.

Competing interests. We declare we have no competing interests.

Funding. This work was supported by the National Institutes of Health (grants nos R00GM114714 and R01AI139154) and start-up funds provided by the University of Kansas.

Acknowledgements. We thank Kelly Dyer, Stuart Macdonald and Amanda Larracuente for providing *Drosophila* strains, and John Kelly, Amanda Larracuente, Ching-Ho Chang and Unckless lab members for consultation on various aspects of the manuscript. We also thank two anonymous reviewers for helpful comments and suggestions on the manuscript.

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
