## [Reviewer comments · Proceedings of the Royal Society B: Biological Sciences]

Review History

RSPB-2019-1534.R0 (Original submission)

Review form: Reviewer 1

Recommendation

Accept with minor revision (please list in comments)

Scientific importance: Is the manuscript an original and important contribution to its field?

Excellent

General interest: Is the paper of sufficient general interest?

Excellent

Quality of the paper: Is the overall quality of the paper suitable?

Excellent

Is the length of the paper justified?

Yes

Should the paper be seen by a specialist statistical reviewer?

No

Do you have any concerns about statistical analyses in this paper? If so, please specify them explicitly in your report.

No

It is a condition of publication that authors make their supporting data, code and materials available - either as supplementary material or hosted in an external repository. Please rate, if applicable, the supporting data on the following criteria.

Is it accessible?

Yes

Is it clear?

Yes

Is it adequate?

Yes

Do you have any ethical concerns with this paper?

No

Comments to the Author

Some of the best studied natural selfish gene drive systems are found in *Drosophila*, including X chromosomes that bias their transmission by interfering with Y chromosome-bearing sperm, and 'Segregation distorter', an autosomal gamete killer in *D. melanogaster*. These are commonly tied up in chromosomal inversions, with interesting evolutionary consequences, including accumulation of deleterious mutations, whose possible fitness effects have not been studied very much.

In this paper, Lea and Unckless test the interesting hypothesis that, possibly as a result of deleterious mutations, chromosomes containing selfish genetic elements in three *Drosophila* species have reduced immune function, by pricking flies with pathogenic bacteria and comparing their survival. They find little evidence for reduced immune function, so this paper reports mostly negative results. One 'Segregation distorter' line did suffer high mortality following infection. The authors follow up on that result by sequencing the dipteracin gene (an antimicrobial peptide with segregating genetic variation that has previously been linked to susceptibility to bacteria) and by comparing dipteracin and cecropin (another antimicrobial peptide) gene expression, but they don't find any consistent differences to explain why that line is especially susceptible.

Despite the mostly negative results, this is a thorough, interesting and well-written paper. The relationship between immune function and meiotic drivers has not been considered before, and this work will likely stimulate more research in other systems and on other fitness effects.

Other comments:

- Methods, page 6: It doesn't look like *D. affinis* SR/ST was infected (not in Fig.1 or in supplemental dataset).

Other minor comments:

- Maybe a stronger title that states the result?

- Results, page 9: spelling of neotestacea on second line of results

- Please add references for *D. neotestacea* carrying Muller A, and *D. affinis*' X having a fusion of Muller A and D (and also on page 14).
- Page 11: and *that* exhibit
- Figure 2: What is X amino acid symbol for Sd-72/CyO dipteracin genotype? Also, the asterisk above 92 is too small.
- Figure 2 legend: Please add the symbol for arginine to make it easier to refer back to the figure.
- Page 14: In an ancestor *of* *D. affinis*
- Page 15: A few thoughts/comments about the last paragraph: I think it would be useful to note that there may be a better chance of uncovering deleterious mutations / immunity costs in systems where SR/SR are not fertile or viable, unlike *D. affinis* and *D. neotestacea*. Also, these SR chromosomes may also be recombining with each other.
- Page 15: Last sentence: Would be useful to mention/add Fuller et al. 2018 biorxiv manuscript that identifies deleterious loss of function mutations in *D. pseudoobscura* SR.
- Page 15: change 'turnover' to 'turn over'

Review form: Reviewer 2

Recommendation

Accept with minor revision (please list in comments)

Scientific importance: Is the manuscript an original and important contribution to its field?

Good

General interest: Is the paper of sufficient general interest?

Acceptable

Quality of the paper: Is the overall quality of the paper suitable?

Good

Is the length of the paper justified?

Yes

Should the paper be seen by a specialist statistical reviewer?

No

Do you have any concerns about statistical analyses in this paper? If so, please specify them explicitly in your report.

No

It is a condition of publication that authors make their supporting data, code and materials available - either as supplementary material or hosted in an external repository. Please rate, if applicable, the supporting data on the following criteria.

Is it accessible?

Yes

Is it clear?

Yes

Is it adequate?

Yes

Do you have any ethical concerns with this paper?

No

Comments to the Author

Please see word document for review. (Appendix A)

Decision letter (RSPB-2019-1534.R0)

20-Aug-2019

Dear Dr Unckless

I am pleased to inform you that your Review manuscript RSPB-2019-1534 entitled "An assessment of the immune costs associated with meiotic drive chromosomes in *Drosophila*" has been accepted for publication in Proceedings B.

The referee(s) do not recommend any further changes. Therefore, please proof-read your manuscript carefully and upload your final files for publication. Because the schedule for publication is very tight, it is a condition of publication that you submit the revised version of your manuscript within 7 days. If you do not think you will be able to meet this date please let me know immediately.

To upload your manuscript, log into <http://mc.manuscriptcentral.com/prsb> and enter your Author Centre, where you will find your manuscript title listed under "Manuscripts with Decisions." Under "Actions," click on "Create a Revision." Your manuscript number has been appended to denote a revision.

You will be unable to make your revisions on the originally submitted version of the manuscript. Instead, upload a new version through your Author Centre.

- 1) A text file of the manuscript (doc, txt, rtf or tex), including the references, tables (including captions) and figure captions. Please remove any tracked changes from the text before submission. PDF files are not an accepted format for the "Main Document".
- 2) A separate electronic file of each figure (tiff, EPS or print-quality PDF preferred). The format should be produced directly from original creation package, or original software format. Please note that PowerPoint files are not accepted.
- 3) Electronic supplementary material: this should be contained in a separate file from the main text and the file name should contain the author's name and journal name, e.g `authorname_procb_ESM_figures.pdf`

All supplementary materials accompanying an accepted article will be treated as in their final form. They will be published alongside the paper on the journal website and posted on the online figshare repository. Files on figshare will be made available approximately one week before the accompanying article so that the supplementary material can be attributed a unique DOI. Please see: <https://royalsociety.org/journals/authors/author-guidelines/>

4) Data-Sharing and data citation

It is a condition of publication that data supporting your paper are made available. Data should be made available either in the electronic supplementary material or through an appropriate repository. Details of how to access data should be included in your paper. Please see <https://royalsociety.org/journals/ethics-policies/data-sharing-mining/> for more details.

If you wish to submit your data to Dryad (<http://datadryad.org/>) and have not already done so you can submit your data via this link <http://datadryad.org/submit?journalID=RSPB&manu=RSPB-2019-1534> which will take you to your unique entry in the Dryad repository.

Once again, thank you for submitting your manuscript to Proceedings B and I look forward to receiving your final version. If you have any questions at all, please do not hesitate to get in touch.

Yours sincerely,
Professor Loeske Kruuk
Editor
<mailto:proceedingsb@royalsociety.org>

Associate Editor
Comments to Author:

Two expert reviewers have read your MS, and both think it is well designed and written. Their comments are pretty minor, although reviewer 2 does suggest a reasonable addition to the discussion, as drive variants are certainly worth discussing. However, I do not think that this will need a major change, a couple of sentences should be enough to cover the point.

Reviewer(s)' Comments to Author:

Referee: 1

Comments to the Author(s)

Some of the best studied natural selfish gene drive systems are found in *Drosophila*, including X chromosomes that bias their transmission by interfering with Y chromosome-bearing sperm, and 'Segregation distorter', an autosomal gamete killer in *D. melanogaster*. These are commonly tied up in chromosomal inversions, with interesting evolutionary consequences, including accumulation of deleterious mutations, whose possible fitness effects have not been studied very much.

In this paper, Lea and Unckless test the interesting hypothesis that, possibly as a result of deleterious mutations, chromosomes containing selfish genetic elements in three *Drosophila* species have reduced immune function, by pricking flies with pathogenic bacteria and comparing their survival. They find little evidence for reduced immune function, so this paper reports mostly negative results. One 'Segregation distorter' line did suffer high mortality following infection. The authors follow up on that result by sequencing the dipteracin gene (an

antimicrobial peptide with segregating genetic variation that has previously been linked to susceptibility to bacteria) and by comparing dipteracin and cecropin (another antimicrobial peptide) gene expression, but they don't find any consistent differences to explain why that line is especially susceptible.

Despite the mostly negative results, this is a thorough, interesting and well-written paper. The relationship between immune function and meiotic drivers has not been considered before, and this work will likely stimulate more research in other systems and on other fitness effects.

Other comments:

- Methods, page 6: It doesn't look like *D. affinis* SR/ST was infected (not in Fig.1 or in supplemental dataset).

Other minor comments:

- Maybe a stronger title that states the result?
- Results, page 9: spelling of neotestacea on second line of results
- Please add references for *D. neotestacea* carrying Muller A, and *D. affinis* X having a fusion of Muller A and D (and also on page 14).
- Page 11: and *that* exhibit
- Figure 2: What is X amino acid symbol for Sd-72/CyO dipteracin genotype? Also, the asterisk above 92 is too small.
- Figure 2 legend: Please add the symbol for arginine to make it easier to refer back to the figure.
- Page 14: In an ancestor *of* *D. affinis*
- Page 15: A few thoughts/comments about the last paragraph: I think it would be useful to note that there may be a better chance of uncovering deleterious mutations / immunity costs in systems where SR/SR are not fertile or viable, unlike *D. affinis* and *D. neotestacea*. Also, these SR chromosomes may also be recombining with each other.
- Page 15: Last sentence: Would be useful to mention/add Fuller et al. 2018 biorxiv manuscript that identifies deleterious loss of function mutations in *D. pseudoobscura* SR.
- Page 15: change 'turnover' to 'turn over'

Referee: 2

Comments to the Author(s)

Please see word document for review.

Decision letter (RSPB-2019-1534.R1)

28-Aug-2019

Dear Dr Unckless

I am pleased to inform you that your manuscript entitled "An assessment of the immune costs associated with meiotic drive chromosomes in *Drosophila*" has been accepted for publication in Proceedings B.

You can expect to receive a proof of your article from our Production office in due course, please check your spam filter if you do not receive it. PLEASE NOTE: you will be given the exact page

length of your paper which may be different from the estimation from Editorial and you may be asked to reduce your paper if it goes over the 10 page limit.

Your article has been estimated as being 8 pages long. Our Production Office will be able to confirm the exact length at proof stage.

Open Access

Paper charges

Sincerely,

Appendix A

Paper title: **An assessment of the immune costs associated with meiotic drive chromosomes in *Drosophila*.**

General comments:

The intention of this paper was to investigate if selfish genetic elements, in this case sperm killing meiotic drivers, carry a cost to individuals that carry them in terms of immune defence capacity. To examine this specifically in the context of sperm-killing meiotic drivers is novel. In addition, the paper spans three different drive systems across 3 species of *Drosophila*, which adds to the value of the paper. Overall, I think this is a useful contribution and researchers interested in the ecology and evolution of natural meiotic drive systems will generally appreciate the paper.

Overall the paper seems well executed technically. However, I am not a specialist in the immune assessment techniques used for evaluating immune expression in *Drosophila melanogaster*. I can therefore only say that it seems to me a sound approach used and I cannot see any major concerns.

I recommend that this paper could be published with minor changes to the majority of the paper and a considerable development of the discussion as outlined below. As there were no line numbers on the paper I have referred to the section where I think a change should be made.

Specific comments/changes

Background: change **'the interaction involves...'** to **'the interaction can involve...'**

Background: change **'two particularly strong selective...'** to **'two strong selective pressures'**

Background: change **'but most documented cases are sex-linked due to the striking resulting phenotype'** to **'but most documented cases are sex linked, likely due to the striking resulting phenotype'**

Background: In the sentence **'It is located on the second chromosome and is associated with inversions that reduce recombination and link *Sd* driver, enhancer and responder loci over large parts of the chromosome.'** Please change *Sd* to *SD*.

Discussion: can you elaborate on **'This is somewhat analogous to the role that epistasis plays in Muller's ratchet, though the conclusions about the effects of epistasis are conflicting [46, 47].'** It is not clear exactly what you mean here.

Wider discussion comments/suggestions/changes

In the background, section when discussing the age of the systems being examined it would be helpful to have some information (if known) about the age of these systems. This seems pertinent if discussing the build-up of mutational load in the chromosomes.

In methods: For survival assays, were flies housed singly for survival or in groups? If housed singly then no problem. If housed in groups then how can you control for flies interacting with each other or being affected by other infected flies.

In discussion: It seems the only species where you have tested multiple version ($n>1$) for driving chromosomes is *Drosophila melanogaster*. Also, the only species where you test multiple driving chromosomes shows differences between them. While much of the previous meiotic drive literature in similar fitness related contexts has been carried out on one SR chromosome vs one ST chromosome,

the fact you find differences as soon as you evaluate multiple versions of an SD driver is both extremely interesting and also warrants some caution when experimenting on single SR vs ST systems. I do not think this detracts hugely from this paper, as it is a phenomenal amount of work to collect, isolate and maintain SR systems from the wild. However, I think more time should be dedicated to addressing this issue, particularly when drive systems can be very old (e.g. *pseudoobscura*), there can be multiple segregating versions of drivers that have different fitness costs (e.g. t-haplotype or SD) and other studies show SR performs variably against different ST chromosome when in competition (e.g. *D. subobscura*).

In discussion: **'An autosomal driver at an equilibrium frequency of 10% would find itself in homozygotes 1% of the time and deleterious mutations could be purged from the drive chromosome. However, there are ample examples of drive chromosomes that, when homozygous, render the carrier inviable or sterile [3, 8, 26].'**

There are ample examples of both drivers that render homozygotes sterile, and those that don't. I would propose a sentence suggesting a comparison between some systems that 'can' recombine and those that definitely can't, would be good answer to whether muller's ratchet could be escaped.